# Pathological mechanism and antisense oligonucleotide-mediated rescue of a non-coding variant suppressing factor 9 RNA biogenesis leading to hemophilia B

Simon Krooss[1,2], Sonja Werwitzke[3], Johannes Kopp[1], Alice Rovai[2], Dirk Varnholt[3], Amelie S. Wachs[1], Aurelie Goyenvalle[4], Annemieke Aarstma-Rus[5], Michael Ott[2], Andreas Tiede[3], Jörg Langemeier[1,6]*, Jens Bohne[1]*

1 Institute of Virology, Hannover Medical School, Hannover, Germany, 2 Department of Gastroenterology, Hepatology and Endocrinology, Hannover Medical School and Twincore Centre for Experimental and Clinical Infection Research, Hannover, Germany, 3 Clinic of Hematology, Oncology and Hemostaseology, Hannover Medical School, Hannover, Germany, 4 Université de Versailles St-Quentin en Yvelines, INSERM U1179, France, 5 Leiden University Medical Center, Leiden, Netherlands, 6 Pediatric Intensive Care Unit, Children's Hospital Bielefeld, Germany

* joerglangemeier@web.de (JL); bohne.jens@mh-hannover.de (JB)

**Data Availability Statement:** All relevant data are within the manuscript and its Supporting Information files.

## Abstract

Loss-of-function mutations in the human coagulation factor 9 (F9) gene lead to hemophilia B. Here, we dissected the consequences and the pathomechanism of a non-coding mutation (c.2545A>G) in the *F9* 3' untranslated region. Using wild type and mutant factor IX (FIX) minigenes we revealed that the mutation leads to reduced *F9* mRNA and FIX protein levels and to lower coagulation activity of cell culture supernatants. The phenotype could not be compensated by increased transcription. The pathomechanism comprises the *de novo* creation of a binding site for the spliceosomal component U1snRNP, which is able to suppress the nearby *F9* poly(A) site. This second, splicing-independent function of U1snRNP was discovered previously and blockade of U1snRNP restored mutant *F9* mRNA expression. In addition, we explored the vice versa approach and masked the mutation by antisense oligonucleotides resulting in significantly increased *F9* mRNA expression and coagulation activity. This treatment may transform the moderate/severe hemophilia B into a mild or subclinical form in the patients. This antisense based strategy is applicable to other mutations in untranslated regions creating deleterious binding sites for cellular proteins.

## Author summary

The elucidation of the pathomechanisms of non-coding variants yields important insights into diseases as well as cellular processes causing the defect. Although these variants may account for the majority of phenotypic variation, only a minority of them can be explained mechanistically. The human coagulation factor 9 3' UTR variant described here converts a non-essential sequence motif into a U1snRNP-binding site with deleterious effects on

**Funding:** This work was supported by the Else-Kröner-Fresenius-Stiftung (2014_A88) to J. B. and the German research foundation (DFG BO2512/6-1) to J. B. and (DFG OT131/6-1) to M. O. S. K. was supported by the Studienstiftung des Deutschen Volkes. The funders had no role in study design, data collection and analysis, decision to publish, or preparation of the manuscript.

**Competing interests:** The authors have declared that no competing interests exist.

RNA 3' end processing at the nearby poly(A) site. Poly(A) site suppression by U1snRNP was described before and it normally protects cellular mRNAs from premature termination. However, if misled by creation of a U1 site close the authentic poly(A) site as in the *F9* 3' UTR, this nuclear surveillance mechanism results in the opposite. Since recognition by U1snRNP depends on sequence complementarity we were able to use antisense oligonucleotides to mask the mutant site and partially restored *F9* mRNA levels. This antisense based strategy may be applicable to other variants in untranslated regions, which create deleterious binding sites for cellular proteins.

## Introduction

Variants in the non-coding part of the genome can result in severe diseases and phenotypes with a high expressivity [1] and the number of variants identifiable by GWAS can explain only a modest fraction of the phenotypic variation [2]. However, the pathomechanism underlying these variants are often difficult to dissect.

Factor IX (FIX) constitutes a serine protease in the enzyme cascade of the intrinsic coagulation pathway. The *F9* gene is located on the X chromosome consisting of 8 exons [3]. Disease-causing variants in the *F9* gene cause complete or partial deficiency of the corresponding coagulation factor, ultimately leading to hemophilia B (OMIM:306900) of varying severity with an incidence of 1 in 25.000 births [4]. The FIX mutation database (http://www.factorix.org;[5] lists 1083 different variants. Of these, 122 are non-coding and four occur in the 3' untranslated region (3'UTR). These non-coding variants can have severe consequences and in addition, they are often part of regulatory sequences, which otherwise would escape our notice [1].

In 1993 a 3'UTR variant (Chr.X, NC_000023.11:g.139563228A>G) was described in four families with moderate to severe hemophilia B [6]. Affected individuals showed FIX activity between <1 and 4.5% of normal. At present, the FIX mutation database lists 21 patients from different geographic regions with this variant (http://www.factorix.org;[7, 8]; S1 Table). The variant may generate a 5' splice site (5'SS) by an A to G transition at position +1157 downstream of the stop codon (c.2545A>G; NM_000133) in the 3' UTR [6]. However, the underlying pathophysiological mechanism remained unsolved.

Eukaryotic pre-mRNAs undergo multiple processing steps. The removal of introns is executed by the spliceosome, which consists of five small nuclear U-rich ribonulceoproteins (UsnRNPs). The 5'SS is recognized by the RNA component of the U1snRNP [9]. The strength of a U1 binding site is primarily determined by its sequence complementarity to U1snRNA [10]. In addition to splicing, mRNAs are cleaved and polyadenylated at their 3' end. Again, several conserved sequence features in close proximity to the hexameric poly(A) signal (PAS) direct the assembly of the cleavage and polyadenylation (CPA) machinery. The CPA is confronted with the task to identify the correct PAS among many close matches. Especially the long mammalian introns harbor many cryptic PAS. Recently, we and others have shown that U1snRNP is also part of a nuclear surveillance mechanism that suppresses cryptic polyadenylation sites and thereby ensures transcript integrity [11, 12].

For the *LAMTOR2* (p14) gene, we demonstrated that a 3'UTR variant creating a U1 binding site leads to PAS suppression, thereby causing an immunodeficiency [12]. Thus, the generation of a *de novo* U1 binding site in the *F9* 3'UTR could have at least two distinct outcomes: aberrant splicing or failure of 3' end processing and RNA degradation. Here, we show that our *F9* variant generates a U1 binding site. Recruitment of U1snRNP does not initiate splicing, but rather leads to suppression of the downstream PAS and to a strong decrease in *F9* mRNA

levels. This phenotype is the primary cause for reduced FIX protein levels and lower coagulation activity. Using different antisense-oligonucleotide based strategies; we rescued the mutant phenotype, which again points to U1snRNP as the key player in the pathomechanism. Furthermore, we opened the venue to a gene therapeutic approach to target such UTR variants. Our study strengthens the importance of non-coding variants as disease-causing and demands their pathomechanistic analysis.

## Results

### The factor 9 variant in the 3'UTR reduces *F9* mRNA expression

The previously described A to G transition at +1157 bp (c.2545A>G) downstream of the *F9* stop codon may create a 5'SS by comparison to the consensus sequence [6]. Based on our results with a similar disease-causing mutation in the *LAMTOR2* gene [12], we thought to revisit this particular variant. The 3'UTR variant is located upstream of the polyadenylation signal (PAS; Fig 1A). Bioinformatic analysis revealed a putative 3'SS downstream of the PAS in a similar distance followed by a second PAS (Fig 1A). Thus, we were left with five possible explanations: i) the variant creates a miRNA binding site; ii) splicing deletes PAS1 with insufficient PAS2 usage; iii) splicing in the 3'UTR more than 50 nts downstream of the *F9* stop codon eliciting non-sense mediated RNA decay (NMD; [13]; iv) the variant creates or deletes a motif for an RNA-binding protein (RBP) and v) the site binds U1snRNP thereby preventing PAS recognition and 3' end processing [14].

First, we generated a minigene consisting of the *F9* cDNA followed by the 3'UTR and downstream genomic sequences (Fig 1A). The mutated sequence shows high complementarity to U1 snRNA with the A to G transition creating the invariant GU of spliceable U2 introns (Fig 1A;[9]). Next, we generated the variant and two other versions either increasing (FIXopt) or decreasing (FIXdown; Fig 1A) sequence identity to the U1snRNA consensus as expressed by the number of possible base pairs to U1snRNA and the MaxEnt score describing splice site strength[15]. The variant creates a high scoring U1 binding site (Fig 1A).

A comparison of *F9* mRNA levels indicates a dramatic decrease of FIX expression in HEK 293T cells transfected with FIXmut (Fig 1B, lanes 3 and4). Thus, the variant exerts its phenotype by downregulating *F9* mRNA amount. Suppression of the FIX PAS causes read-through into the plasmid backbone (Fig 1B and 1D). Notably, neither optimization (FIXopt; Fig 1B, lane 5) nor deoptimization (FIXdown; Fig 1B, lane 2) of the sequence did alter *F9* mRNA levels in a significant manner compared to the mutant or the wild type, respectively. To analyze this in more detail and to definitely exclude miRNAs as causative factors, we cloned another version containing the variant combined with low complementarity to U1snRNA (mutdown, S1A Fig). Although this construct carries the variant, the inability to bind U1 prevents RNA decrease (S1B Fig). This demonstrates the relevance of the mutated position for U1snRNP recognition transforming a sequence motif into a strong U1 binding site causing FIX RNA decrease.

The *F9* gene holds a potential 3'SS 198 nts downstream of its PAS (Fig 1A). This can give rise to a splicing event initiated from the newly created 5'SS and thereby removing the canonical PAS leading to nonsense-mediated RNA decay [13]] or to a general failure in 3' end processing due to an insufficient PAS2. To examine these possibilities, a 3'Rapid Amplification of cDNA Ends (3'RACE) was conducted (Figs 1A and S1C). Sequencing indicates the absence of a splicing event, since the gene-specific forward primer is located upstream of the potential 5'SS (S1C Fig). In addition, the 3'RACE revealed appropriate polyadenylation at the annotated site in both FIXwt and FIXmut transfected cells (Fig 1C upper and lower panel). Importantly, these results show that the residual, non-read-through mutant *F9* mRNA (Fig 1B, lane 4) still underwent 3' end processing at the *F9* PAS1. Polyadenylation at PAS2 occurred neither in the wildtype and nor in the mutant. To excluded splicing by a more direct approach we performed

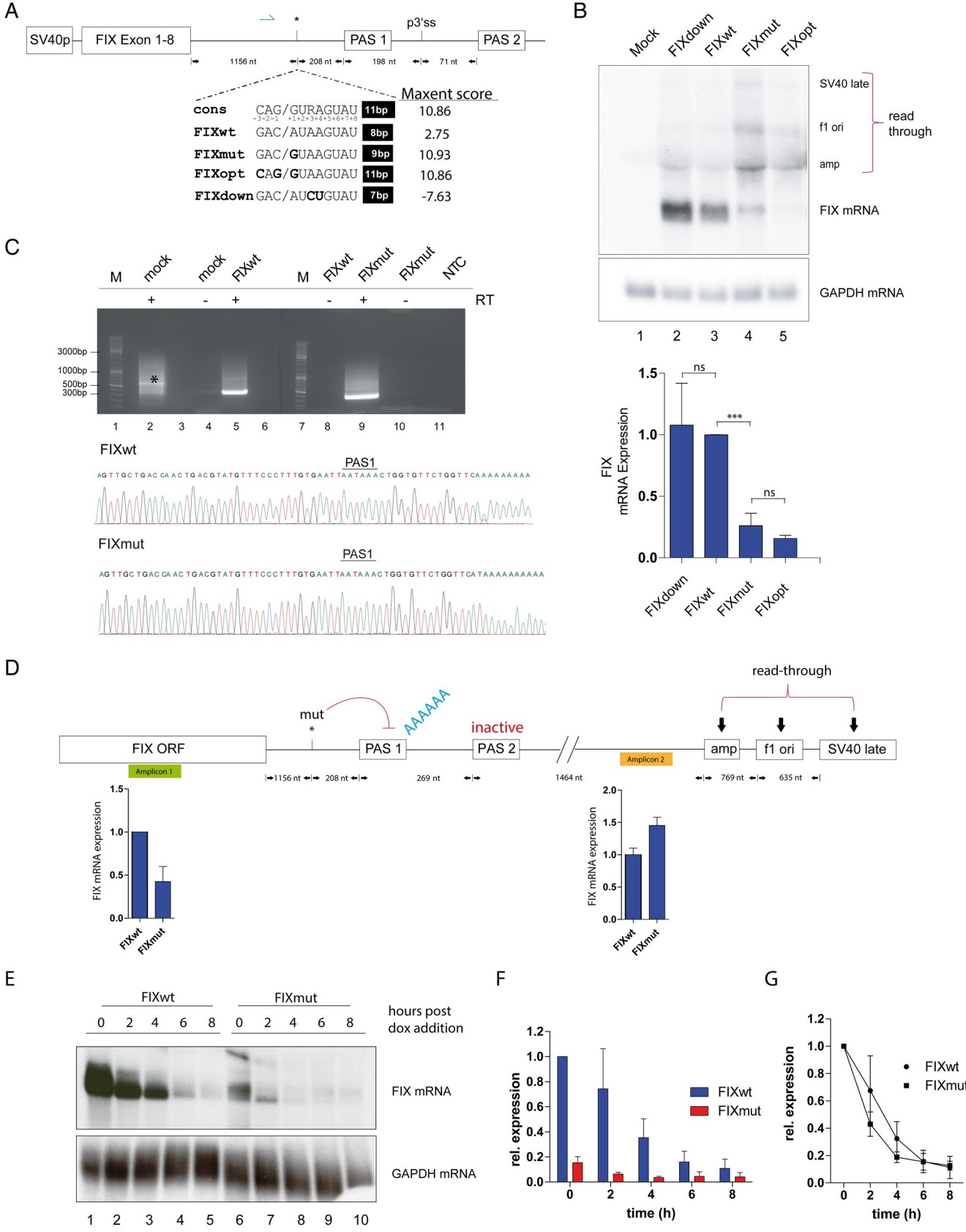

**Fig 1. The variant leads to lower *F9* RNA levels and creates a putative 5' splice site.** A) Depiction of the SV40 driven Factor IX minigene. The *F9* cDNA (Exon 1–8) is followed by its authentic 3'UTR. The position of the primer used for the 3'RACE is marked by a blue arrow. The variant is indicated by an asterisk at position 1156 downstream of the stop codon and 208 nts upstream of the polyadenylation site 1 (PAS1). Further downstream a putative 3'SS (p3'ss) as well as a potential second polyadenylation site (PAS2) was annotated. Another 421 nts of genomic sequence downstream of PAS2 were included. The sequence of the putative U1 binding site (5'SS) (FIXmut) is opposed to the corresponding regions of the wildtype and the 5'SS consensus sequence. The oblique outlines the exon/intron border within a common 5'SS. The numbers of possible base pairs to U1 snRNA are shown on the right, followed by the respective MaxEnt score. The position of each nucleotide within the consensus sequence is indicated by numbering. In addition, we cloned minigenes harboring either an optimized (FIXopt) or no U1 binding site (FIXdown). B) Northern blot using total RNA obtained from HEK 293T cells transfected with the indicated constructs. The membrane was hybridized with a $P^{32}$-labelled probe corresponding to the *F9* cDNA. The position of *F9* mRNA is indicated on the right. Glyceraldehyde 3-phosphate dehydrogenase (GAPDH) serves as a loading control. Bands were densitometrically quantified using a phosphoimager (lower panel). The wildtype was set to 1. The standard deviation represents 3 independent experiments. An unpaired student's T-test was performed (ns, non-significant; *** p<0.0001). To monitor transfection efficiency all experiments included transfection of a eGFP encoding plasmid. C) Agarose gel electrophoresis of 3'RACE products. Two gels are shown with markers on the left side. The presence of reverse transcriptase (RT) during cDNA synthesis is indicated (+/-). The band around 300bp represents the amplicon with the forward primer shown in (A). The signal marked with an asterisk in the untreated sample (mock) represents an amplicon of the MLX gene. In the lower panel Sanger sequencing of *F9* specific amplicons showed polyadenylation at the correct *F9* PAS1. D) Schematic of *F9* ORF, 3'UTR and downstream sequences including ampicillin sequence, f1 ori and SV40 poly (A) signal. Suppressive effect of the variant (marked by an asterisk) on PAS1 is indicated in red. Green box symbolizes the qRT-PCR amplicon 1 detecting all *F9* RNA species and the yellow box indicates amplicon 2 for measuring *F9* RNA species derived from read-through. qRT-PCR results are depicted below the corresponding amplicons. Each qRT-PCR experiment was conducted in biological and technical triplicates. E) Tet-off *F9* RNA half-life experiment. pTetbi FIXwt and pTetbi FIXmut plasmids were transfected into Hela TA cells. 36 hours post transfection, doxycycline was added to shut off transcription. Total RNA was collected 0, 2, 4, 6 and 8 hours post shut off and analyzed by Northern blot (n = 3). F) Quantification of *F9* RNA obtained from (E). FIXwt was set to 1 and all FIX bands were GAPDH normalized. G) Depiction of relative *F9* RNA decay rate, both FIXwt and FIXmut were set to 1 to allow direct comparison.

RT-PCR with primers located upstream of the potential 5'SS and downstream of the putative 3'SS. Again, no splicing could be detected (S1D Fig). Thus, the lack of splicing from the new, cryptic 5'SS excludes NMD and processing at PAS2 as pathomechanisms.

The PAS inhibition exerted by U1 is only transient and spatially limited to roughly 1 kb [16, 17] and allows processing at downstream sites as shown in Fig 1B. To quantify the extent of read-through we used qRT-PCR with one amplicon detecting all *F9* mRNA and a second measuring only the read-through (Fig 1D). The variant increased read-through (amplicon 2, Fig 1D, right side) as already observed in the Northern blot analysis (Fig 1B). The downregulation of mutant *F9* mRNA is not as pronounced, because all RNAs including the read-through are detected (amplicon 1, Fig 1D, left side).

The residual RNA expression observed in Fig 1B and 1D could be an artifact of the plasmid-based transient transfection. Thus, we introduced our FIX minigene into a pre-tagged, doxycycline-inducible locus to obtain a stable, single copy *F9* HeLa cell line for the wild type and the mutant (S1E Fig). Even from this genomic localization a residual RNA expression for the mutant was detectable (S1E Fig).

The observed reduction in *F9* mRNA could be due to co-transcriptional RNA decay as described for suppression of 3' end processing [18]. Alternatively, the variant may alter the *F9* mRNA stability post-transcriptionally. To analyze this, we assessed the steady-state level of *F9* mRNA and its half-life. Using a doxycycline inducible version of the FIX minigenes, we shut off transcription 36 hours post transfection. The wild type *F9* mRNA produced from this minigene decays with a half-life of approximately 3 hours (Fig 1E and 1F). The mutant *F9* mRNA is expressed at lower levels already at time point zero (compare to Fig 1B). The half-life analysis suggests only a slightly accelerated decay (Fig 1G). Thus, the mutation does not alter the half-life of *F9* mRNA.

## A modified FIX minigene phenocopies poly(A) site suppression on the protein level

Our initial attempts to detect FIX protein and to measure active FIX in the supernatant of several cell lines transfected with our minigenes failed due to low expression levels (Fig 2B, lanes 2 and 3). The usage of the CMV promoter and the 5'UTR of pcDNA3.1 (Fig 2A) slightly enhanced the *F9* mRNA expression, but still recapitulated the phenotype of the variant (Fig

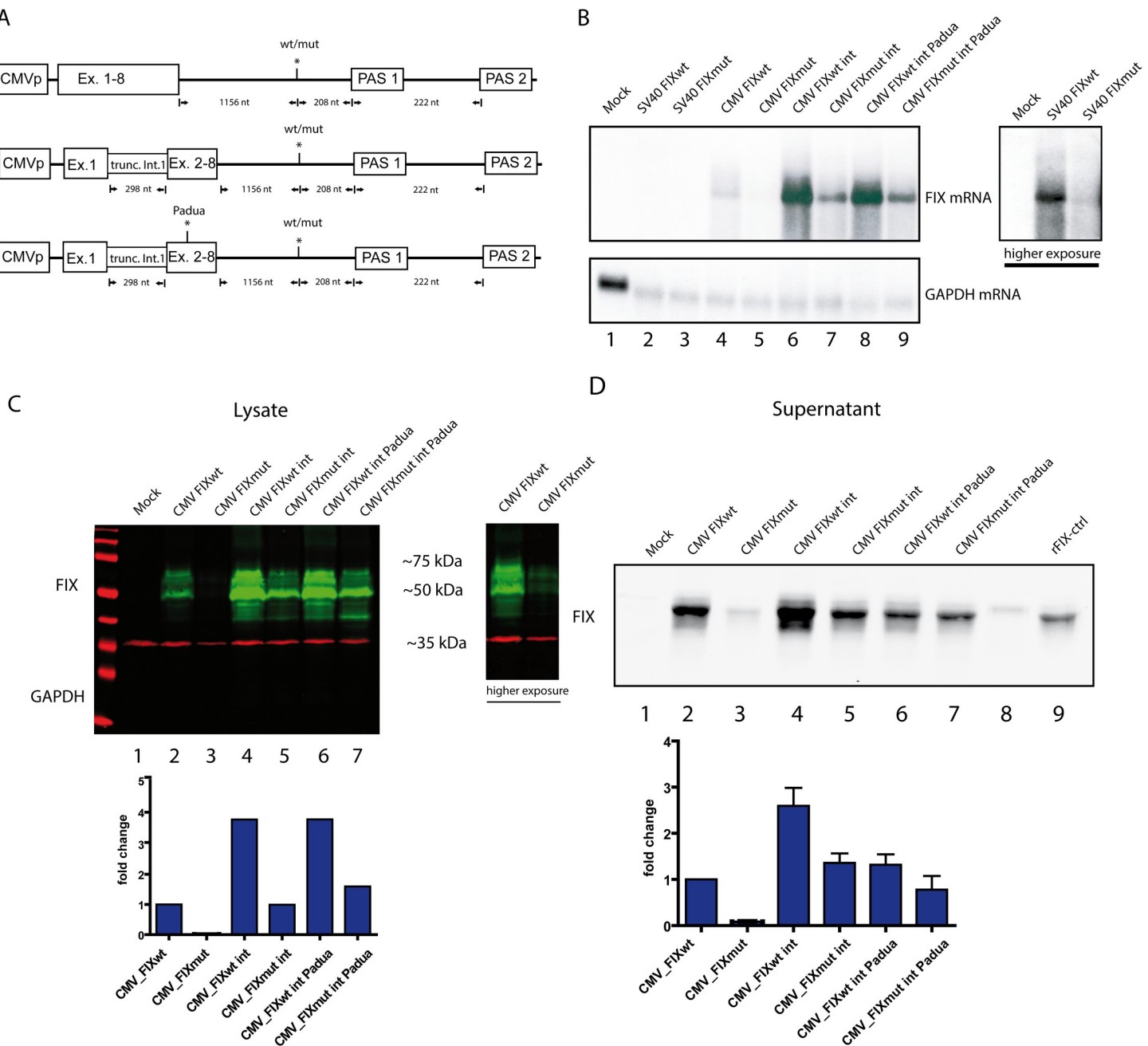

**Fig 2. Establishment of a FIX protein expression system. A)** Vector architecture of three different CMV driven FIX minigenes with or without 3'UTR variant. Two constructs (middle and lower) were equipped with a truncated FIX intron 1 located in between exon 1 and exon 2. The lower construct combines truncated intron 1 and the gain-of-function mutation denoted as "Padua". **B)** Northern blot using total RNA of HEK 293T cells transfected with the indicated constructs. GAPDH served as a loading control. A higher exposure of lanes 2 and 3 is shown in the right panel. **C)** Western blot of HEK 293T cell lysate transfected with the respective constructs visualized by Licor Odyssey and two antibodies labelled with different fluorophores (FIX in green). GAPDH (red) serves as loading control. Cells were lysed 30 hours post transfection. A higher exposure of lanes 1 to 3 is shown in the right panel. A densitometric quantification is depicted in the lower panel. The CMV-driven FIXwt construct was set to 1. **D)** Immunoblot analysis of supernatant collected from transfected HEK 293T cells with the respective constructs at 30 hours post transfection. The densitometric analysis was performed as in C. Standard deviations represent three independent experiments.

2B, lanes 4 and 5). Now we observed FIX protein both in whole cell lysates (Fig 2C, lanes 2,3 and inset) and in the supernatant (Fig 2D, lanes 2 and3). Since FIX is synthesized in hepatocytes, we sought to verify our results in hepatic cell lines. To this end we transfected the human

hepatoma cell line Huh7 with the CMV-driven FIX constructs (S2 Fig). Also, in this cell line the variant leads to 4-fold downregulation of *F9* RNA (S2A Fig). The same is true for FIX protein in cell culture supernatants (S2B and S2C Fig).

We additionally inserted a truncated *F9* intron 1 with its authentic splice sites at the correct position into the *F9* cDNA (Fig 2A, middle panel;[19]). The intron is efficiently spliced (S3A Fig) and it strongly enhances *F9* mRNA and protein levels (Fig 2B, lane 6 and 2C, lane 4). Elevated transcription rather than splicing seems a likely explanation for this increase, since incorporation of the terminal *F9* intron did not lead to higher mRNA levels (S3B–S3D Fig) and work done by the Kurachi laboratory also implied the presence of a transcriptional enhancer [19]. Of note, the variant again led to lower RNA and protein levels even though the basal expression was higher (Fig 2B, lane 7 and 2C, lane 5). This argues for a high expressivity of the variant and a tight suppression of 3' end processing. Finally, we took advantage of a previously published gain-of-function variant referred to as Padua [20] (Fig 2A, lower panel) yielding a hyperactive FIX. Since the Padua mutation exerts its effect on FIX protein activity, the mRNA levels remained unchanged (Fig 2B, lanes 8 and 9). However, protein levels in the supernatant were slightly reduced (Fig 2D, lanes 6 and7). Since FIX is highly modified on the post-translational level [21], we included recombinant FIX as a size-matched control showing a higher molecular weight (Fig 2D, lane 9) indicating insufficiently modified FIX protein in both the cell lysate and the supernatant (Fig 2C and 2D). Densitometric analysis revealed that the mutant phenotype gets blurred in the context of FIX intron and the Padua mutation (Fig 2D, lower panel). However, in the cell lysates a robust 4-fold decrease was observed on protein level (Fig 2C, lower panel) comparable to the reduction of *F9* mRNA (Fig 1B, lower panel).

## The improved FIX minigene phenocopies the mutant also in reduced coagulation activity

CHO cells were shown to produce active FIX *in-vitro* to levels almost comparable with primary hepatocytes and they are harnessed to purify FIX for injection into hemophilic patients [22]. And indeed, transfection of the minigenes into CHO cells could reproduce the phenotype at the level of RNA (Fig 3A) and intracellular FIX protein (Fig 3B, lanes 4,5 and 6,7). Transfection of CHO cells displayed a favourable ratio of modified vs. unmodified FIX protein (compare Figs 2C and 3B). In addition, we quantified FIX antigen (FIX:Ag) by ELISA and observed a 4-fold reduction for all mutant constructs (Fig 3C). To assess coagulation activity, we employed of a one-stage coagulation assay [23]. Again a 4-fold decrease in activity was observed, when comparing mutant with wildtype constructs (Fig 3D). This was also observed in hepatic Huh7 cells albeit on a lower level (S2D Fig). The Padua mutation displayed a clear gain-of-function phenotype as described previously (Fig 3D; [20]). Although protein and FIX: Ag levels were quite similar (Fig 3B and 3C), the activity was enhanced by 10 fold (Fig 3D). Again, the phenotype of our variant persisted in the hyperactive Padua context showing that the variant acts on a wide range of activity levels (Fig 3D).

In summary, the non-coding variant creates a U1 binding site in close proximity to the *F9* poly(A) site and this results in failure of 3' end processing and ultimately *F9* mRNA degradation. The decreased mRNA levels result in lower protein amounts, less secreted FIX and finally reduced FIX antigen and coagulation activity, thus partially reflecting the clinical phenotype observed in patients [24].

## Blockade of U1snRNP rescues FIX expression

To demonstrate that the binding of U1snRNP to the mutated region is responsible for the decrease in *F9* mRNA, we blocked U1snRNP using Antisense Morpholino Oligonucleotides

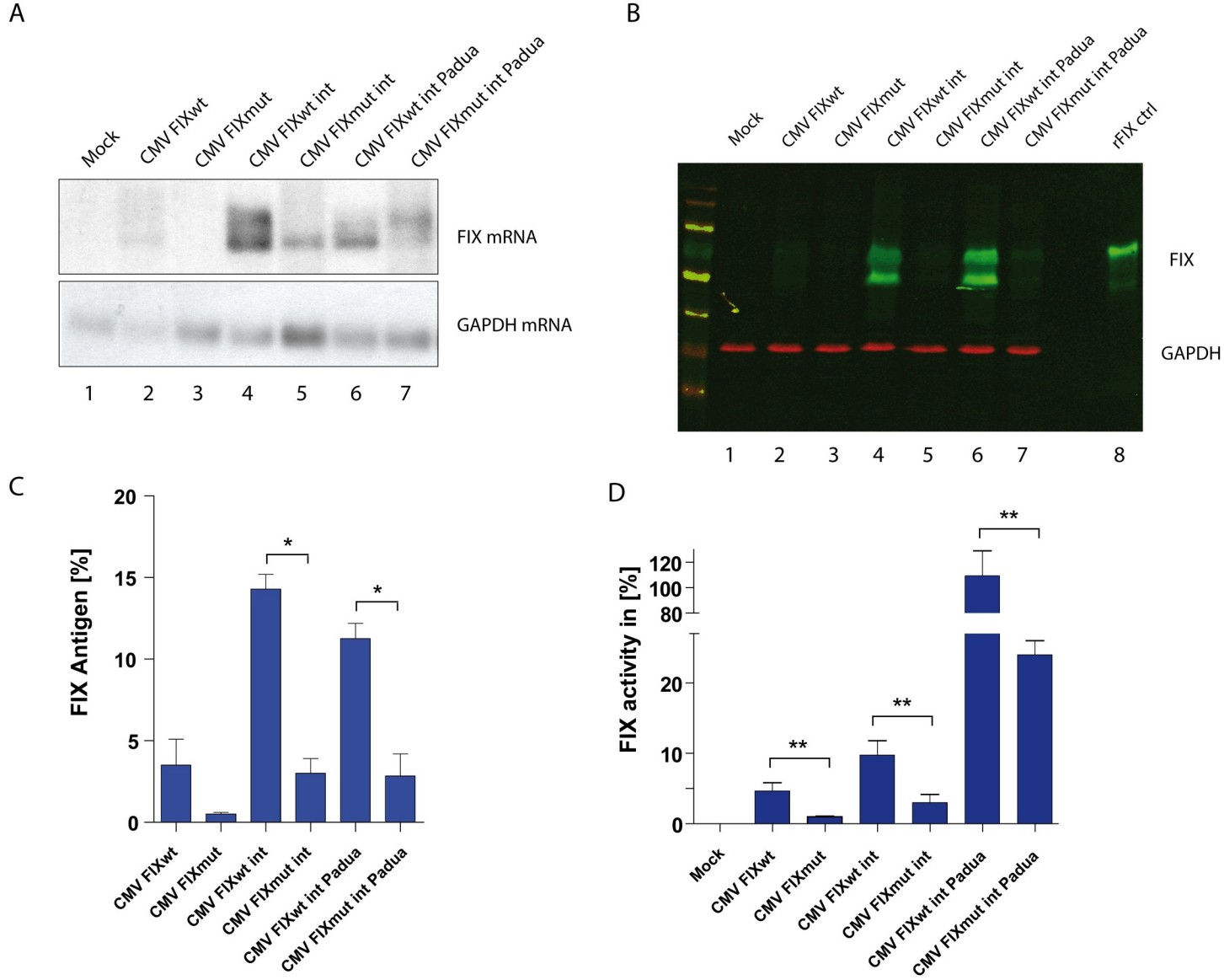

**Fig 3. The *F9* 3'UTR variant reduces FIX antigen and coagulation activity. A)** Northern blot of CHO total RNA collected 30 hours post transfection with CMV driven FIX minigenes. GAPDH serves as a loading control. **B)** Western blot of CHO cell lysate 30 hours post transfection with the respective minigenes. GAPDH was used as a loading control and recombinant FIX serves as a positive control (Lane 8). **C)** Supernatant of FIX transfected cells collected 96 hours post transfection was subjected to quantification using a FIX ELISA (n = 3). Significant differences determined via Bonferroni's Multiple Comparison Test are indicated by an asterisk *(p<0.05)*. **(D)** Coagulation activity of FIX transfected CHO supernatant was determined 96 hours post transfection (n = 3). Asterisks indicate statistical differences *(p<0.05)*.

(AMOs) targeting U1snRNA (Fig 4A, red sequence). U1 AMOs block 5'SS recognition by binding to the free U1 snRNA 5' end and additional nucleotides thereby invading and disrupting the secondary structure of U1snRNA [16]. We transfected the respective FIX minigenes followed by introduction of the AMOs in HeLa cells since they proved to be more resilient to the treatment. The mutant phenotype remained unaltered in the presence of a control AMO (Fig 4B, lanes 2,3 and lowest panel). The transfection of U1-specific AMOs facilitated a strong increase in mutant *F9* mRNA expression by 3-fold (Fig 4B, lane 5 and lowest panel). Thus, the blockade of U1snRNP rescues the *F9* mutant mRNA expression. Interestingly, both FIXwt and FIXmut minigenes showed elevated expression (Fig 4B, lanes 4,5). However, the level of

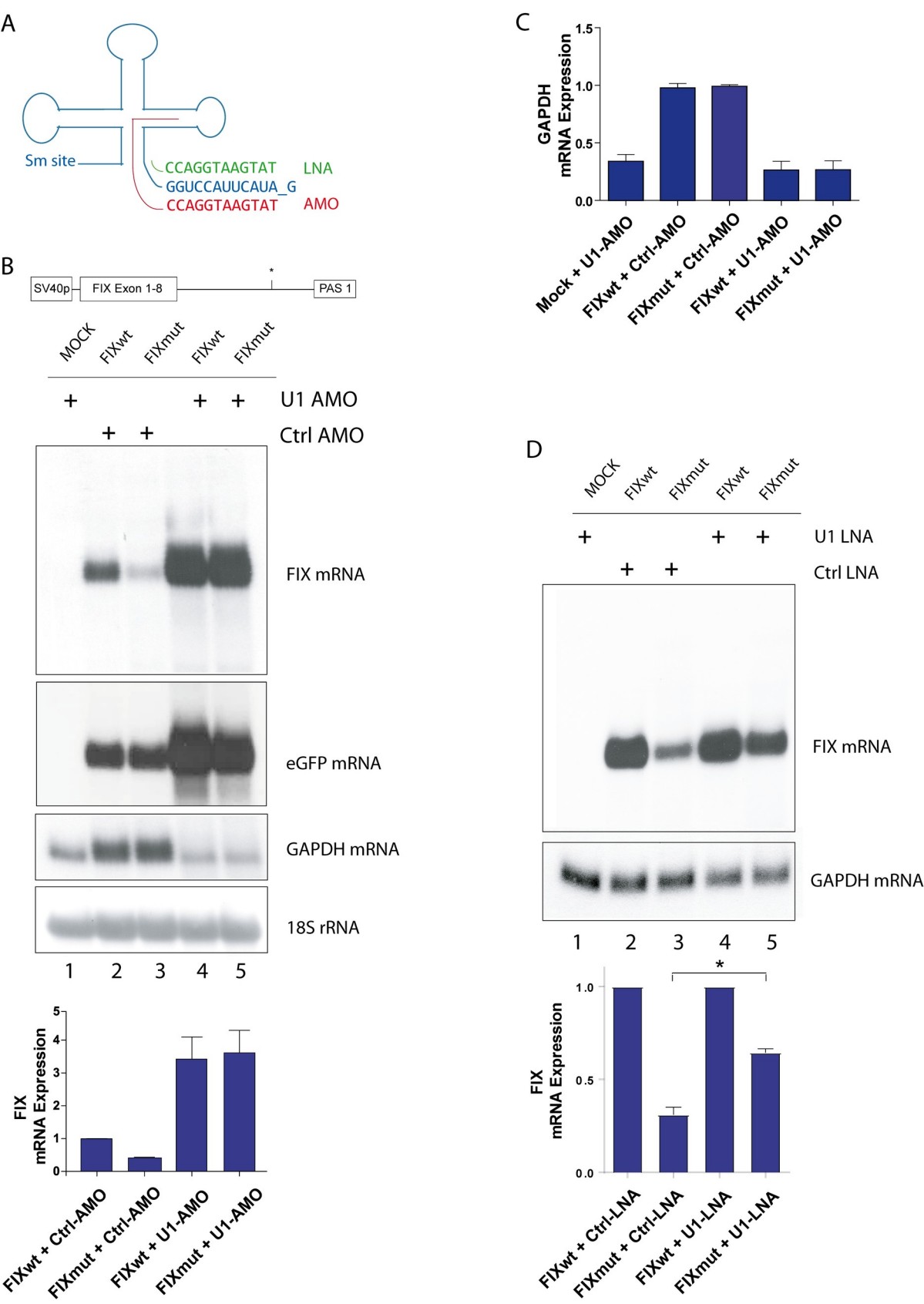

**Fig 4. Blockade of U1snRNP rescues mutant _F9_ RNA expression. A)** U1snRNA and its 5'SS recognition sequence (blue). Binding properties of locked nucleic acids (LNAs), indicated in green and antisense morpholino oligonucleotides (AMOs), indicated in red. **B)** Depiction of SV40-driven F9 expression construct. Below, Northern blot using total RNA from FIXwt/mut transfected Hela cells performed as in Fig 1B. Control or anti U1 AMOs were transfected 14 h post FIX minigene transfection as indicated. The position of FIX RNA is marked on the right. The blot was re-hybridized with a GAPDH-specific, an eGFP-specific and an 18S rRNA-specific probe. Densitometric analysis of FIX mRNA expression normalized to 18S RNA is shown on the lowest panel. The FIXwt in presence of control AMOs was set to 1 (n = 3). **C)** Densitometric analysis of GAPDH mRNA expression. GAPDH mRNA of FIXmut transfected cells plus control AMOs was set to 1. **D)** Northern blot as performed in (B) except that control and anti U1 LNAs were used. The statistical analysis of three independent experiments is shown below. The asterisks indicate that the partial rescue due the blockade of U1 is significant as determined via Student's T-test.

GAPDH mRNA was decreased due to splicing inhibition upon U1snRNP blockade (Fig 4B). The 3-fold decrease in GAPDH mRNA (Fig 4C) strongly correlated with its mRNA half-life in HeLa cells of 8 hours [25]. To rule out loading differences we took advantage of 18S rRNA as an unspliced, pol I derived transcript (Fig 4B). Interestingly, expression from a GFP-encoding plasmid is also increased by anti U1 AMOs (Fig 4B). Thus, the reason for enhanced FIX expression in the wildtype seems to be a general effect of the U1 AMOs on intron-less genes. Very recently U1 AMOs were shown to increase RNA pol II density at transcriptional start sites, which may explain our observation [26].

In a second approach, a blockade of U1snRNP has been performed using locked nucleic acids (LNAs) designed exclusively against the U1snRNA 5' end (Fig 4A, green sequence), thereby retaining the U1snRNA secondary structure. Masking of the U1snRNA 5' end led to a rescue of _F9_ mRNA expression in FIXmut transfected cells (Fig 4D, lanes 4 and 5), whereas cotransfection with a control LNA still showed reduced FIXmut mRNA expression. The rescue was incomplete due to the lower transfection efficiency of LNAs vs. AMOs. Again, the GAPDH expression was reduced due to the manipulation of splicing. Importantly, after LNA treatment the wildtype _F9_ mRNA expression remained constant (Fig 4D, lanes 2 and 4).

## A U1snRNA suppressor mutant reduces wild type F9 RNA expression

In a third approach we sought to co-express a suppressor U1snRNA, which is designed to bind the wildtype sequence and to a lesser extent to the variant (Fig 5A; [12]). To this end we co-transfected our FIX minigenes along with the U1 suppressor mutant plasmid or an empty plasmid as a control. The retargeting of the suppressor U1snRNP to the wildtype minigene decreases _F9_ mRNA expression (Fig 5B). RNA expression from the mutant minigene remains constant (Fig 5B). A rescue of the mutant RNA by the suppressor U1 snRNA is not expected since a G•U base pair still allows binding of the suppressor to mutated sequence (Fig 5A and 5B). In addition, high levels of endogenous U1snRNP are still repressing 3' end processing of mutant FIX minigene.

This approach together with the data presented above clearly demonstrates a direct involvement of U1snRNP in the suppression of 3' end processing in the context of the 3'UTR variant. In addition, we only found one miRNA binding site, which seed region overlaps with the variant but leading to a lower score (S4 Fig). Moreover, we employed the algorithm RBP map [27] to identify whether the variant changed binding motifs for RBPs. The variant, however, creates an ANKHD1 binding site, which is involved in signaling [28] and transcriptional regulation via histone binding [29] and is thus unlikely to be responsible for the observed phenotype.

## Masking the mutated site rescues FIX expression

While the mechanistic approach clearly showed that the pathomechanism of the variant depends on U1snRNP recognition, blockade of U1 is not a therapeutic option. We followed a vice versa approach and aimed to mask the newly created U1 binding site by chemically

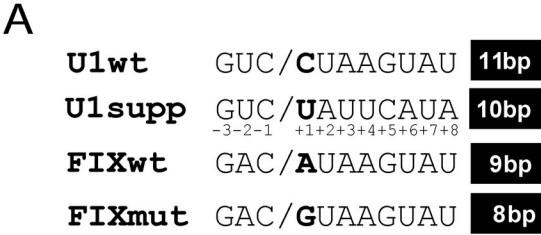

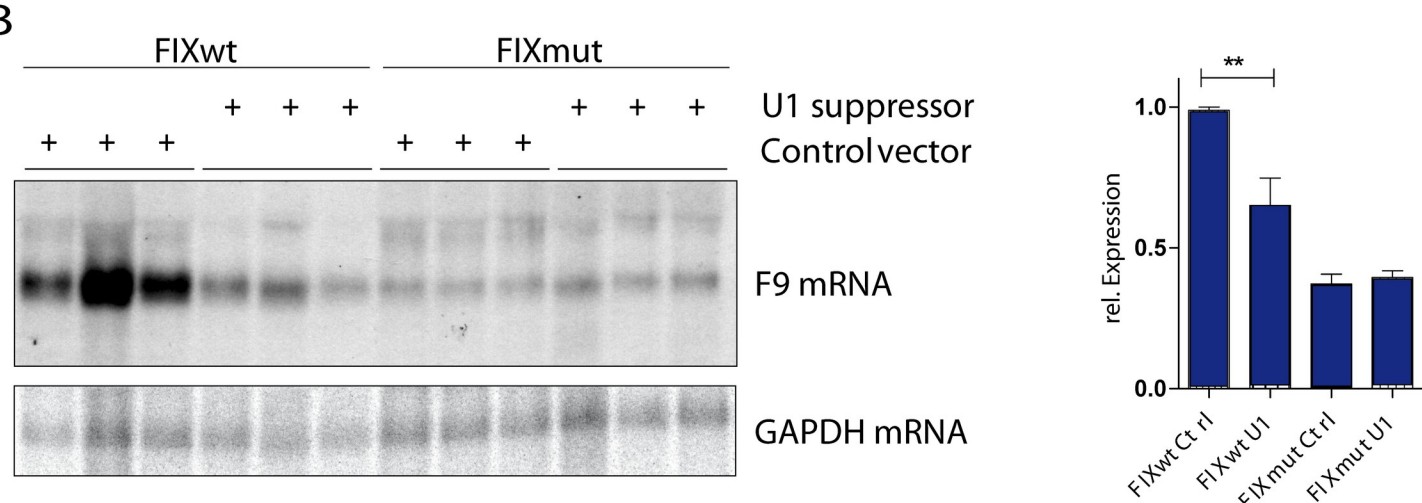

**Fig 5. A U1 snRNA suppressor mutant reduces wild type F9 RNA expression.** (A) The U1 suppressor mutation is depicted. In addition, the wildtype sequence (FIXwt) is opposed to the corresponding regions of the variant (FIXmut) and the 5'SS consensus sequence. The oblique outlines the exon/intron border. The number of possible base pairs to U1snRNA are shown on the right. The position of each nucleotide within the consensus sequence is indicated by numbering. (B) HEK 293T were transfected in triplicates with the indicated minigenes and U1 suppressor plasmids or control vectors. The Northern blot was hybridized with a F9 specific probe and re-hybridized with a GAPDH probe for normalization. A statistical analysis of three independent experiments is shown on the right side. ** = p<0.004.

modified RNA antisense oligonucleotides (2OMePS) displaying increased stability [30] and therapeutic potential in Duchenne muscular dystrophy (DMD;[31, 32]. We designed a 20mer 2OMePS antisense oligonucleotide hybridizing to both the wildtype and the mutated sequence (Fig 6A). To this end we transfected our original SV40-driven FIX minigenes along with different concentrations of control and targeting oligonucleotides into HeLa cells, which again are more resilient to the 2OMePS treatment (Fig 6B). This vector change was necessary, since our optimized constructs produced high amounts of *F9* mRNA, which could not be rescued due to a lower efficiency of 2OMePS transfection and thus, a lower nuclear antisense oligonucleotide concentration in HeLa cells. While in transfections with a control oligo the decrease of mutated *F9* mRNA persisted (Fig 6B, lane 2 and 3), masking the site to prevent association of U1snRNP leads to a rescue of mutant *F9* mRNA expression (Fig 6B, lane 4 and 5). Higher concentrations of the oligo exhibited some toxicity as shown by the lower GAPDH signal (Fig 6B, lane 6). Interestingly, an additional *F9* mRNA species is detectable in both wildtype and mutant context indicating a switch to a processing site beyond PAS2 (Fig 1A).

The same strategy can also be applied using a viral vector derived system. Here, the snRNP system is 'hijacked' by replacing the 5' part of natural snRNAs with an antisense sequence of choice. This strategy protects the antisense sequence of choice from degradation. U7snRNA is often used for this purpose, which normally ensures histone RNA processing (Fig 6C;[33, 34].

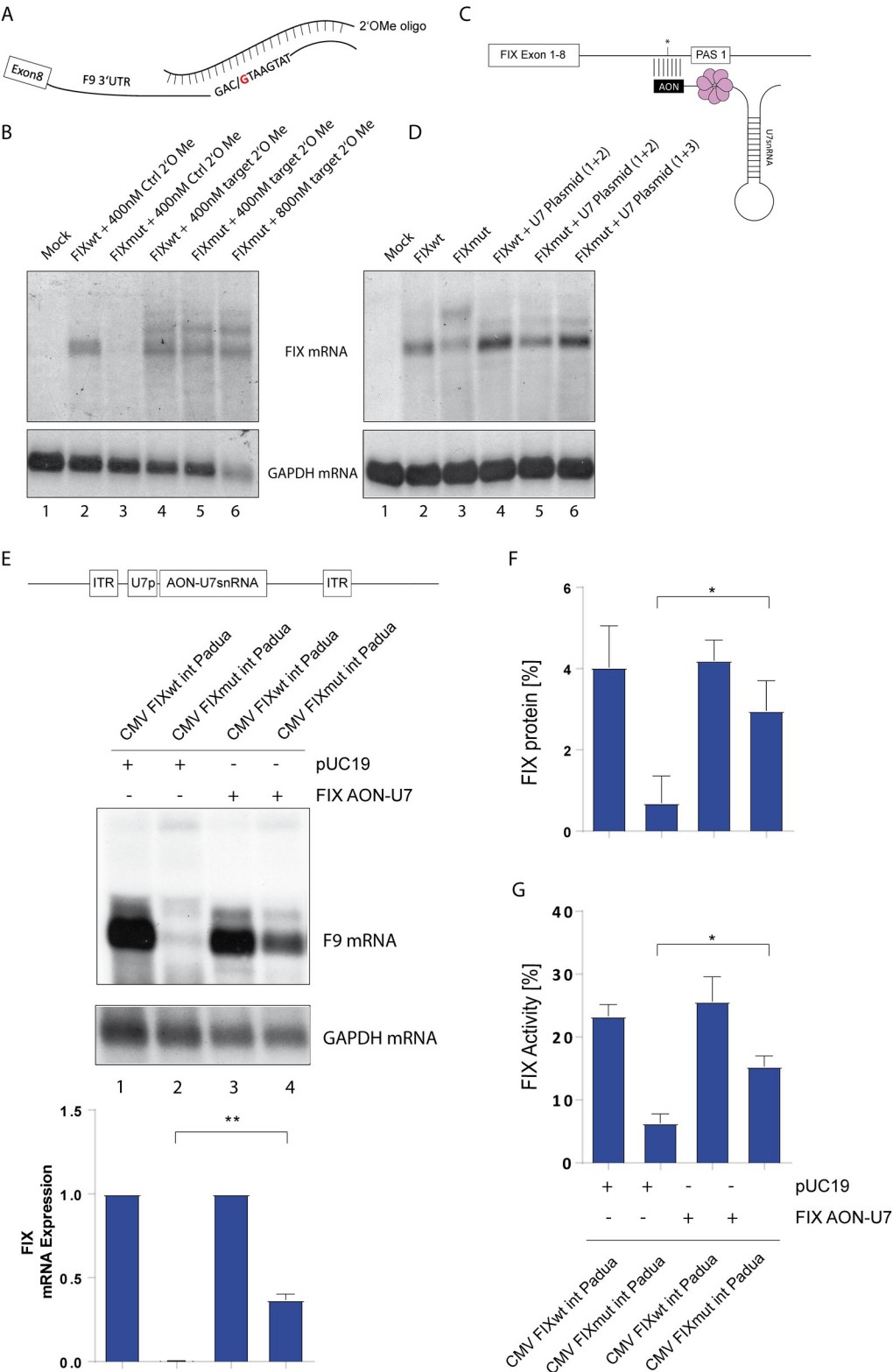

**Fig 6. Masking of mutated region via two different antisense oligonucleotides. A)** Depiction of the mutationally generated U1 binding site being masked by a complementary 2'-O-Methyl oligonucleotide (2OMePS). **B)** Northern blot of total RNA after co-transfection of SV40 driven FIXwt and mut constructs with control or targeting 2OMePS oligonucleotides. Hela cells were harvested 16h post transfection. GAPDH serves as a loading control. **C)** Depiction of

the modified U7snRNP targeting the mutated region on the FIX mRNA thereby masking the U1snRNP binding site. **D)** Northern blot of total RNA after cotransfection of SV40 driven FIX minigenes and U7-antisense expression vector in indicated molar ratio (1+2/1+3). Hela cells were harvested after 16 hours as in (B) and GAPDH serves as a loading control. **E)** Depiction of the AAV vector backbone, which was used in transient transfections (top). The inverted terminal repeats (ITR) delineate the vector-derived sequences. A cassette consisting of the U7 promoter (U7p) and a fusion of the FIX antisense sequence (AON) to U7 snRNA was inserted. The lower panel the RNA analysis by northern blot using total RNA from transfected CHO cells in presence of the AAV plasmid expressing the antisense oligonucleotide fused to U7snRNA or pUC19 as a control. GAPDH serves as a loading control. Below three independent experiments were quantified by phosphoimager analysis. The partial rescue in the presence of the antisense oligonucleotide is indicated by a double asterisk *(p = 0.0025)*. **F)** Supernatant of FIX transfected CHO cells collected 96 hours post transfection was subjected to quantification using a FIX ELISA (n = 3). Significant differences determined via Bonferroni's Multiple Comparison Test are indicated by an asterisk *(p = 0.05)*. **G)** Coagulation activity in percent matched to a standard curve of recombinant factor IX. The tested supernatants were derived from CHO cells transfected with the indicated plasmids. Standard deviations represent three independent experiments. The asterisks indicate a statistically significant elevation of FIX activity *(p<0.05)*.

We constructed a plasmid based on an adeno-associated virus (AAV) vector backbone [35, 36] and inserted a longer antisense sequence of 40 nucleotides complementary to the *F9* 3'UTR fused to U7snRNA (Fig 6C and 6E; upper scheme). This sequence shows no hits in a Blast search (S4B Fig). Transfection along with the FIX minigenes showed a rescue of mutant *F9* mRNA expression (Fig 6D, lanes 4–6). Again, a more downstream PAS is used when the U1 binding site is masked (Fig 6D, lane 6). Next, we introduced the FIX minigene and the U7 plasmid into CHO cells by transfection and monitored *F9* mRNA by Northern blot (Fig 6E, middle panel). The co-transfection of the U7 plasmid led to a significant increase in mutant *F9* mRNA expression (Fig 6E, lowest panel). This rescue is also detectable on the protein level in the supernatant measure by FIX ELISA (Fig 6F) and finally also the coagulation activity was increased by blocking U1 access to the mutant site (Fig 6G). The absolute activity was lower compared to Fig 3 due to altered molar ratios of the transfected FIX plasmid (Fig 6G). This confirms the potential of our strategy to be applied as a gene therapeutic approach to treat non-coding and disease-causing variants leading to hemophilia B in this case and other diseases.

## Discussion

A reassessment of UTR variants in *F9* revealed a point mutation possibly creating a 5'SS [6]. We promisingly recapitulated the phenotype demonstrating a downregulation of *F9* mRNA in the presence of the variant. A key player of this pathomechanism is U1snRNP, which erroneously binds within the 3'UTR and leads to a PAS suppression resulting in poor mRNA expression. Thus, the variant misleads the U1-dependent surveillance mechanism, which normally suppresses intronic poly(A) sites [37] and a failure of 3' end processing results in RNA degradation[18].

The *F9* mRNA repression also translates into reduced FIX protein levels explaining the hemophilia B observed in the patients (http://www.factorix.org; [8]). While we clarified the pathomechanism, our minigene system is not able to perfectly mirror the reduction of FIX antigen and coagulation activity observed in the patients. This is mainly due to a higher residual mutant RNA expression. In addition, the read-through RNAs (Fig 1B and 1D) may also contribute to FIX protein, thus slightly compensating the effect of the variant. Also, a human hepatic cell line displayed the same residual RNA expression (S2 Fig) and so did a single integrated copy of *F9* under control of a dox-inducible promoter (S1C Fig). We speculate that the genomic location on the X chromosome in conjunction with local regulatory sequences is the reason for this discrepancy. In addition, we always assayed for accumulation of FIX in the supernatant compared to secretion and circulation/consumption of FIX in the patients.

Nevertheless, our *F9* minigene proved to be a useful tool to uncover the pathomechanism of the non-coding variant.

Despite the different levels of transcription, the phenotype of the variant persisted, which again stresses its high expressivity also shown by the reduced coagulation activity of our mutant supernatants. At the molecular level this can be explained by the high excess of U1snRNP in the nucleus leading to a tight suppression of 3' end processing. Due to issues on FIX modifications and activity during *in-vitro* experiments, we sought to use the Padua mutation as a "magnifier" to thoroughly test the expressivity of our non-coding variant. Now the coagulation assay reached the same activity as recombinant FIX and the 4-fold downregulation of the variant persisted also in this context.

We reasoned that shielding of the mutant site would prevent U1snRNP-binding and alleviate PAS suppression. Two types of antisense oligonucleotide (ASO) strategies previously demonstrated efficacy in other models: i) 2OMePS RNA oligos [32, 38] with some side effects reported [39] and ii) an plasmid based system expressing an antisense oligo fused to U7 snRNA [35]. In essence, both antisense strategies lead to a rescue of *F9* mRNA expression and an increase in coagulation activity, which may be able to revert the phenotype into a mild or subclinical hemophilia B. Interestingly, the RNA duplex formation and association with U7snRNP did not hamper translation in the cytosol, which suggests detachment of the complex at the nuclear pore or during the pioneer round of translation. However, we will have to determine if the partial rescue was due to limitation of the cellular processing capacity of snRNP particles [40].

Most hemophilia B patients suffer from disease-causing variants in the coding region (http://www.factorix.org). Some of these patients have been treated by gene-addition therapy introducing a non-mutated *F9* cDNA as it has been done using adeno-associated viral vectors (AAV) in clinical trials [41]. We constructed similar vectors expressing the antisense-U7snRNA cassette as mentioned above and produced recombinant AAV particles. However, cell culture experiments using AAV serotype 2 did not reach sufficient antisense RNA levels to reproduce the partial rescue of FIX activity observed by transfection of the AAV vector plasmid (Fig 5E). We speculate that the U7 hairpin structure may serve as a resolving signal for the AAV rep protein leading to the aberrant packaging of truncated genomes. Currently we are optimizing the vector design. In general, this current study might be a starting point for dissecting and treating other UTR variants, which exert their effects by misplacing cellular RNA-binding proteins on the mutant RNAs. These variants may cover a wide range of phenotypes from modulation of RNA half-life to mislocalization in the cytosol. For example, the Factor VIII mutation database list four 3'UTR variants. One advantage of the variant masking strategy compared to gene addition is the ongoing physiological expression of the mutated gene as only transcribed RNAs are targeted and rescued.

## Materials and methods

### Expression vectors

The human *F9* cDNA (NM_000133.3) was cloned into pGL3 (Promega). The 3'UTR including the variant and 597 bp of downstream sequence was synthesized by MWG operon.

FIXwt was restored by PCR mutagenesis (see primer list in Table 1). FIXdown and FIXopt were generated by PCR with primers P9, P17 and P18 and cloned into pGL3 using EcoRI and XbaI. The 5'UTR of pcDNA3.1 and a CMV promoter were inserted using BglII and HindIII. A sequence comprising Factor IX Padua and a truncated *F9* intron 1 was synthesized by MWG operon and inserted into the CMV driven factor IX minigene via HindIII and BamHI. The Padua mutation has been replaced by the authentic sequence using BstBI and MfeI in a partial

digest. The vector coding for U7snRNA linked to a mutated factor IX complementary region descends from pSMD2 U7dtex23 [42]. *F9* specific antisense nucleotides were inserted into the U7snRNA gene by three PCR reactions (P8+P10, P9+P11 and P10+P11) and implemented into pSMD2 GFP via XbaI/NheI. *F9* vectors employed for the decay experiments and stable cell lines are derived from CMV FIXwt int Padua or CMV FIXmut int Padua. *F9* cDNA sequence and 3'UTR were amplified using P16+P17 and transferred into a pTetbi promoter driven vector (kindly provided by V. Cordes, MPI Göttingen) via AgeI, BamHI and NcoI digest. FIX mutdown was generated via PCR mutagenesis using primers P1, P18, P19 and P20. Products of PCR1 (P1+P18) and PCR2 (P19 and P20) were employed for a fusion PCR using P1 and P20. The corresponding PCR product was cloned into FIXwt via EcoRI/XbaI, thereby replacing the wild type sequence and giving rise to FIX mutdown. Cloning of the U1 suppressor mutant was done by modifying the 5'SS recognition site of the U1snRNA to bind „GACA-TAAGT". This modified U1snRNA including the authentic U1snRNA promoter was synthesized (IDT-DNA) and cloned into the U1 wt vector [12] via HindIII/BamHI.

## Cell culture and transfection

HEK 293T, HeLa and CHO cells were grown in DMEM supplemented with 10% fetal calf serum, 1mM sodium pyruvate, and 1% penicillin/streptomycin and 4μg/mL vitamin K at 37°C and 5% $CO_2$. One day before transfection, $5\times10^6$ 293T cells were seeded in a 10-cm dish to a confluency of 70–80%.

Transfections were performed using calcium phosphate and a total DNA amount of 15.5μg (5μg FIX plasmid, 0.5μg eGFP DNA and 10μg of pUC19 DNA). Cells were harvested 16h post transfection for RNA analysis. pTetbi FIXwt or pTetbi FIXmut were co-transfected with a Flp-recombinase vector into Hela-EM2-11ht cells [43]. Single cell clones were evaluated via Mammalian beta-Galactosidase Assay Kit (Thermofisher) according to the manufacturer's protocol. Half-life experiments were conducted in HeLa TA cells harboring a stably integrated copy of the Tet-transactivator by the addition of 20 ng Doxycyclin. For U1 blockade, HeLa cells were transfected with FIXwt and FIXmut using Viafect (Promega). 18 hours post transfection, cells were PBS-washed and trypsinised. After centrifugation, $10^6$ cells were resuspended in 50μL of Buffer "R" (Neon Transfection System, Life Technologies) and 2μL of 300nmol of antisense morpholino oligonucleotides (AMOs) were added. The cell suspension was successively exposed to two pulses of 35ms width and an input voltage of 1005 V. After electroporation, cells were transferred in 2mL of antibiotic-free medium and incubated overnight before harvesting. For U1 blockade via locked nucleic acids (LNAs) $5\times10^5$ HeLa cells were seeded in a six-well. A total DNA amount of 2μg was transfected using Viafect (Promega). 4 hours post transfection, 10nmol of anti U1 or control LNAs (Exiqon) were transfected using Lipofectamine 3000. Cells were harvested 20 hours post LNA transfection. For the U1 suppressor experiments, HEK 293T cells were seeded into 6-well plates. At a confluency of approximately 70%, cells were transfected with 0.7ug FIXwt or FIXmut and 1.8ug of either U1 suppressor vector or empty plasmid (pUC19) using Lipofectamine 3000. RNA was harvested 30 hours post transfection and subjected to Northern blot analysis.

For FIX activity analysis, $4\times10^5$ CHO cells were seeded in a six-well. At a confluency of about 50%, cells were transfected with various FIX minigenes using Viafect (Promega). 12 hours post transfection, medium was replaced with FCS-free DMEM, supplemented with 4μg/mL vitamin K, 2.5% BSA and 1% L-Glutamine and cells were incubated for 96 hours.

For the 2OMePS mediated rescue, $5\times10^5$ HeLa cells were seeded into a six-well one day before transfection. A total DNA amount of 2μg was transfected using Viafect (Promega). 4 hours post viafection, 400nmol of targeting (2OMePS) or fluorescent control oligo were

**Table 1. List of oligonucleotides.**

| Cloning of | P1 (EcoRI for) | 5'-GAGGAAGAATTCAACAGTGTGTCTTCAGC-3' |
|---|---|---|
| *F9* cDNA | P2 (FIX_opt_rev) | 5'-GGAGACATGATACTTACCTGCTCTGGTC-3' |
| | P3 (FIX_down_rev) | 5'-GGAGACATGATACAGATGTCCTCTGGTC-3' |
| Cloning of | P4 (FIX_ovl_DS_for_wt) | 5'-GTTGAAGTTGCCTAGACCAGAGGACATAAG-3' |
| *F9* downstream | P5 (FIX_ovl_US_rev_wt) | 5'-GTATGCTAGTTAAAGGAGACATGATACTTATGTCCTC-3' |
| Sequences | P6 (3'UTR BamHI for) | 5'-GCGGATCCTGAAAGATGGATTTCCAAGGTTAATTCATTG-3' |
| | P7 (FIX_ovl_flank_DS_rev) | 5'-GAGCATTGAGAAAGCGCCACGC-3' |
| Cloning of | P8 (U7-FIX-Reverse) | 5'-AGAGGACGTAAGTATCATGTCTTTGCGGAAGTGCGTCTGTA-3' |
| U7 construct | P9 (U7-FIX-Forward) | 5'-TTACGTCCTCTGGTCTAGGAATTTTTGGAGCAGGTTTTCT -3' |
| | P10 (FU7-Xba) | 5'-GGGTCTAGATAACAACATAGGAGCTGTGA-3' |
| | P11 (RU7-Nhe) | 5'-AAAGCTAGCCACAACGCGTTTCCTAGGA-3' |
| Amplicon 1 | P12 (*F9*_ORF_qPCR_for) | 5'-ATTCCTATGAATGTTGGTGTCCCT-3' |
| | P13 (*F9*_ORF_qPCR_rev) | 5'-GGGTGCTTTGAGTGATG TTATCCAA-3' |
| Amplicon 2 | P14 (*F9*_readthrough_for) | 5'-GTGCACCTATAATCCCAGCTACTGGGGAG-3' |
| | P15 (*F9*_readthrough_rev) | 5'- CAGTCATAAGTGCGGCGACG-3' |
| Stable cell line | P16 (cl_*F9*NcoI_for) | 5'- GCGCGCCATGGCAAGCTTACCACTTTCACAATCTGCTAGC-3' |
| | P17 (cl-*F9*AgeI_rev) | 5'-CGCGACCGGTGCGAATTGGGATGCCTCTCCATG-3' |
| FIXmutdown | P18 (FIX mut down rev) | 5'-GGAGACATGATTAAGACGACCTCTGGTC-3' |
| | P19 (FIX mut down for) | 5'-GTCGTCTTAATCATGTCTCCTTTAAC-3' |
| | P20 (Xba rev) | 5'- ACTCTCTAGAAGATTCAAGATAGAAG-3' |
| | P21 (cr ds primer) | 5'- CTGGGCCCAGCCAAGAAATTTAA-3' |
| Antisense | AMO oligonucleotide—Control | 5'-CTTCTTACCTCAGTTACAATTTATA-3' |
| Morpholinos | AMO oligonucleotide–anti U1 | 5'-GGTATCTCCCCTGCCAGGTAAGTAT-3' |
| LNAs | Anti U1 Locked Nucleic Acid | 5'-G+CC+AG+GT+AA+GT+AT-3' |
| | Control Locked Nucleic Acid | 5'-G+CC+AA+CT+CA+CT+AT-3' |
| 2OMePs | 2'O Methyl anti FIX oligo | 5'-UGAUACUUACGUCCUCUGGU-3' |
| Oligos | 2'-O-Methyl control oligo | 5'-AAAAGAAAACAUUCACAAAAUGGG-3' |

transfected using 5µL of Lipofectamine 3000 (Invitrogen). RNA was harvested 16 hours after second transfection. U7 mediated rescue was performed via cotransfection of SV40 driven FIXwt or FIX mut and pSMD2-FIX-U7 into HeLa cells in a molar ratio of 1:3 using viafect. Medium was changed after 4 hours and 16h post transfection; cells were harvested for RNA isolation.

## RNA preparation and analysis

RNA methods were performed as described previously [12]. For detection of *F9* RNA, a specific probe corresponding to the *F9* cDNA was generated from the FIX plasmid by HindIII/BamHI digestion. The GAPDH-specific probe was prepared as described [12]. cDNA synthesis for quantitative PCR (qPCR) was conducted using QuantiTect Reverse Transcription Kit (Qiagen) according to the manufacturer's protocol. Prior to reverse transcription, RNA was treated with TURBO DNase (Invitrogen) and purified via RNeasy columns (Qiagen). qPCR was performed with QuantiTect SYBR Green PCR Kit (Qiagen) using primer pairs P12+P13 (amplicon 1) and P14+P15 (amplicon 2). For 3'RACE experiments, poly(A)+ -RNA of transfected cells was isolated and 50 ng were reverse transcribed. To this, the Invitrogen 3'-RACE protocol and the provided reagents were utilized. The PCR products were separated via agarose gel electrophoresis. After gel extraction, the DNA was transferred into pCR2.1 (Invitrogen) for sequencing.

## Western blot analysis and factor IX activity measurement

For detection of FIX protein, the F9 monoclonal antibody (M01), clone 2C9 (Abnova) was employed in a dilution of 1:500. GAPDH protein expression was detected using GAPDH antibody (1:2000; Cell Signaling Technology). Detection was performed via licor Odyssey infrared imaging system. Recombinant FIX (BeneFix, Pfizer) was employed as positive control. Supernatant obtained from CHO cells was employed for ELISA-based quantification of FIX antigen (FIX:Ag) using ZYMUTEST Factor IX from HYPHEN BioMed according to the manufacturer's protocol including a standard FIX calibrator concentration set to 100% which equals 100 U/dl. Thus, FIX:Ag levels are expressed as % FIX protein. FIX activity was determined using a one-stage coagulation assay. Samples were diluted 1:5 in imidazol buffer and mixed with an equal amount of FIX deficient plasma and Actin FS APTT reagent (all from Siemens Healthcare). After incubation at 37˚C for 2 min, 0.025 M calcium-chloride was added, and the coagulation time recorded in an Amelung KC10 coagulometer. Calibration curves were generated with human standard plasma diluted in FIX deficient plasma.

## Bioinformatic analysis and statistics

MicroRNA binding sites were analyzed using the online prediction tool miRDB [44]. RBPmap [27] is available here: http://rbpmap.technion.ac.il/. Splice site prediction was performed using NNSPLICE 0.9 version [45] and MaxEnt score [15]. Statistical analysis (Bonferroni's Multiple Comparison test) was performed using the GraphPad Prism software.

## Supporting information

**S1 Fig. Mutational analysis and stable FIX cell line.**
(PDF)

**S2 Fig. Analysis of the phenotype in the hepatic cell line HuH-7.**
(PDF)

**S3 Fig. Splicing of the first truncated and the terminal FIX intron.**
(PDF)

**S4 Fig. miRNA analysis and BLAT screening for off-target effects.**
(PDF)

**S1 Table. Clinical and phenotypic data of hemophilia B patients carrying the same mutation.**
(PDF)

## Acknowledgments

We thank Thomas Schulz for ongoing encouragement.

## Author Contributions

**Conceptualization:** Andreas Tiede, Jörg Langemeier, Jens Bohne.

**Formal analysis:** Sonja Werwitzke, Dirk Varnholt, Amelie S. Wachs.

**Funding acquisition:** Jens Bohne.

**Investigation:** Simon Krooss, Jörg Langemeier, Jens Bohne.

**Methodology:** Johannes Kopp, Alice Rovai, Dirk Varnholt, Andreas Tiede, Jörg Langemeier.

**Project administration:** Jens Bohne.

**Resources:** Sonja Werwitzke, Amelie S. Wachs, Aurelie Goyenvalle, Annemieke Aarstma-Rus, Michael Ott, Andreas Tiede, Jens Bohne.

**Supervision:** Jörg Langemeier, Jens Bohne.

**Validation:** Jens Bohne.

**Visualization:** Jens Bohne.

**Writing – original draft:** Simon Krooss, Jörg Langemeier, Jens Bohne.

**Writing – review & editing:** Sonja Werwitzke, Dirk Varnholt, Amelie S. Wachs, Aurelie Goyenvalle, Annemieke Aarstma-Rus, Michael Ott, Andreas Tiede, Jörg Langemeier, Jens Bohne.

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
