## [Decision Letter · Decision Letter 0]

8 Oct 2019

Dear Dr Bohne,

Thank you very much for submitting your Research Article entitled 'Pathological mechanism and antisense oligonucleotide-mediated rescue of a non-coding variant suppressing factor 9 RNA biogenesis leading to hemophilia B' to PLOS Genetics. Your manuscript was fully evaluated at the editorial level and by independent peer reviewers. The reviewers appreciated the attention to an important problem, but raised some substantial concerns about the current manuscript. Based on the reviews, we will not be able to accept this version of the manuscript, but we would be willing to review again a much-revised version. We cannot, of course, promise publication at that time.

If you decide to revise the manuscript for further consideration at PLOS Genetics, please aim to resubmit within the next 60 days, unless it will take extra time to address the concerns of the reviewers, in which case we would appreciate an expected resubmission date by email to plosgenetics@plos.org.

[LINK]

We are sorry that we cannot be more positive about your manuscript at this stage. Please do not hesitate to contact us if you have any concerns or questions.

Yours sincerely,

Gregory M. Cooper, PhD

Associate Editor

PLOS Genetics

Gregory Barsh

Editor-in-Chief

PLOS Genetics

Reviewer's Responses to Questions

**Comments to the Authors:**

Reviewer #1: In this paper, the authors describe their examination of the mechanism of a specific variant in the 3’ UTR of the factor 9 (F9) gene that is known to cause hemophilia B. After presenting several potential mechanisms, they pursue the hypothesis that the variant creates a U1snRNP binding site that in turn leads to inappropriate binding and loss of 3’ end processing. Through the use of various synthetic F9 wild-type and mutant expression constructs, the authors demonstrate reduction of F9 mRNA levels corresponding with the presence of the variant. Additionally the authors demonstrate that U1snRNP is necessary for this mRNA reduction, partially rescuing the phenotype when U1snRNP is inhibited. Lastly, they present data from ASOs targeting the variant itself, again demonstrating partial rescue and thereby suggesting a potential therapeutic strategy.

Overall, the paper presents a convincing argument that the variant’s deleterious effect is through U1snRNP. They present the results of many well-controlled experiments that combine to create a consistent picture of U1snRNPs role. My main concern is the use of several different cell contexts. It seems that HEK 293T, CHO, Huh7, and HeLa cells are all used in different places. It is disappointing that better data was not obtained in the liver-derived Huh7 cells that might be conjectured to be the most relevant to factor 9 biology. It appears that some cell types were better suited for protein assays and others for mRNA assays. The authors must clarify the reasons for using different cell types in the various assays. Nevertheless, I believe with these changes and a few others this manuscript is suitable for publication in PLOS Genetics.

Some minor issues:

1. p.3 line 82: “At present, the FIX mutation database lists 21 patients from different geographic regions (http://www.factorix.org; (7, 8); supplementary table 1)”. I believe “with this variant” should be added to the end of the sentence.

2. p.3 line 91: “Again several conserved sequence features the hexameric poly(A) signal (PAS) direct the assembly of the cleavage and polyadenylation (CPA) machinery.” This is confusing and should be re-worked.

3. p. 7 line 191: “The intron is efficiently spliced (Fig.192 S3A) and it strongly enhances F9 RNA and protein levels (Fig. 2B, lane 6 and 2C, lane 4).” Is there evidence of an enhancer element in this intron?

4. P. 10 line 269 “This vector change was necessary, since our optimized constructs are highly overexpressed in HeLa cells making rescue experiments difficult (data not shown)” This is a bit concerning. It ties into my general concern about the switching of cell types throughout the paper.

Reviewer #2: The authors have investigated the mechanism by which a noncoding mutation in the 3’UTR of the Factor IX (FIX) serine protease of the coagulation pathway causes disease. The nucleotide substitution was described in 1993 as a cause of moderate to severe hemophilia B with FIX activity between <1 and 4.5 % of normal. The mutant site resembled a 5’ splice site suggesting a cryptic splicing event. The authors propose that while the resemblance to a 5’ splice site recruits the U1 snRNP, the effect is not splicing but suppression of polyadenylation that is another function of U1. They use antisense oligos that bind to the mutant site to block binding of U1 and rescue polyadenylation. The authors provided reasonable evidence that the mutation does not induce use of a cryptic 5’ splice site and that in a plasmid minigene the mutation causes reduced polyadenylation leading to readthrough transcription in the plasmid. This conclusion is also supported by qRT-PCR. Antisense oligos (ASOs) that bind U1 and block interactions with 5’ splice sites and the mutant site rescue expression of FIX. ASOs and a U7 snRNP directed block of the mutation in the FIX 3’UTR provided a partial rescue. One major concern is that all work was done using transfected, or rarely, integrated minigenes. Minigenes with different architectures gave similar results but still it would be particularly convincing if the results were shown in patient cells if they are available. The results are a proof of principle but given the partial rescues in Fig 5 and the reliance on only minigenes, the results are borderline preliminary.

The results in Figs 2 and 3 could be combined and there is redundancy with the use of different minigenes

Fig 2D – clarify “supernatant”. Presumbably this is protein from the media but not clear

Using and ASO to block U1 is an unusual strategy since there should be an effect on most splicing. However here the results support the contention that the mutation affects U1 binding rather than a microRNA.

While the rescue in Fig 5 was significant, there was not as strong as effect as one might expect. Why weren’t AMOs rather than 2OMe antisense oligos were used to block the mutant site on the minigene mRNA since the AMOs worked so well on U1 snRNA? We are number of ASOS tested? Perhaps there are secondary structures of other features that prevent sufficient interaction of ASOs with the mutant site.

Reviewer #3: In this interesting mns, the Authors explore the pathological consequence of a previously reported mutation associated to hemophilia B and located in the 3’UTR of the F9 gene. Through the identification of an U1 snRNP-mediated polyA suppression mechanism that reduce the amount of F9 transcripts the Authors developed an antisense oligonucleotide rescue strategy to correct this type of defect.

The pathological consequences of mutations in the 3’UTR are difficult to study and not always obvious. Here the Authors provide convincing evidence with reporter systems that a F9 mutation in the 3’UTR affect the amount of F9 transcripts through the creation of novel U1 consensus binding site that possibly supress the polyA. The Authors have previously reported a mutation with a similar disease causing-mechanism in another gene (REF 12) and here they provide novel evidence that this type of defects can be corrected by a novel ASO-based therapeutic strategy. Even if the effect of the mutation on transcript and F9 activity as well as the ASO therapeutic approach are clear and convincing, I feel that some additional experiments should be performed to better clarify the mechanistic connection between the U1 and polyA site suppression.

Major points.

1- The Authors do not completely exclude an effect on splicing. It is essential to prove that the mutation creates a functional 5' splice site (SS) and that its recognition by the spliceosomal component U1 snRNP causes F9 mRNA suppression in the absence of splicing. Specifically, (Page 6 lane from lane 149), the 3’RACE experiments do not exclude that the mutation induces a splicing-dependent removal of the first polyA site. This splicing event might occur through the usage of the upstream mutant-created 5’ss and the cryptic downstream 3’ss. To clarify this aspect, the Authors should perform a RT-PCR experiment with primers located before and after the two sites to see if this region is spliced. More in general this part is confused and difficult to follow and should be completely rewritten (for example fig S1C does not seem to evaluate “readthrough” but only the expression levels).

2-To have a more direct evidence that U1 is involved in polyA suppression (and not splicing), I feel that the Authors should necessarily perform experiment with suppressor U1s that bind to the WT region as previously done in ref 12 and show that the suppressor U1s reduce expression. This is important as the polyA site and the U1 binding site seems quite distant (208 nucleotides).

In addition, the formal prove that the U1 affect polyA is missing as they just show less transcript and “read through”. The experiments I suggest at point 1 and 2 will be useful to reinforce the main mechanistic conclusions that U1 directly suppresses polyA.

4- Some conclusions are overestimated. In the absence of specific experiments, the conclusions should be toned down. I provide here a list of sentences that, in my opinion, need attention.

- Page 11 lane 303. I do not feel that the experiments performed really prove that the variant affect the nuclear quality control and that a failure of 3’ end processing result in RNA degradation. I suggest to tone down/ explain better in light of the available experimental evidence provided.

-Page 12 lane 336. The evidence that the ASO binding to the mutant site induce a switch to the downstream polyA site is not evident from the experiment and should be better explained.

-Page 5 lane 149. As the mutation is in the non-coding 3’UTR, I do not understand why the splicing removal of the first polyA site should induce NMD.

Minor points

- Page 3 lane 88. I think that the snRNPs involved in splicing are five and not six. (U7 is in 3’ end processing of histones)

-The rationale of using the gain of function Padua mutant in the context of the 3’UTR variant is unclear, as it seems simply an additional control.

-The Authors claim that transcription is not involved? Why increased transcription does not compensate? This is strange.

-Page 4 lane 112. I think that the most appropriate verb is “reduces” and not “impedes” F9 expression

-The Authors identified with in silico approaches two polyA sites that were then tested in expression plasmids and the effect of the mutation is on the proximal one. Is this polyA site the most used or the only one used also in vivo in hepatocytes?

-Fig 1 B and S1B lack of appropriate statistical analysis. Fig 3b lacks quantification.

**Have all data underlying the figures and results presented in the manuscript been provided?**

Reviewer #1: Yes

Reviewer #2: Yes

Reviewer #3: Yes

PLOS authors have the option to publish the peer review history of their article (what does this mean?). If published, this will include your full peer review and any attached files.

Reviewer #1: No

Reviewer #2: No

Reviewer #3: No

---

## [Decision Letter · Decision Letter 1]

3 Feb 2020

Dear Dr Bohne,

Thank you very much for submitting your Research Article entitled 'Pathological mechanism and antisense oligonucleotide-mediated rescue of a non-coding variant suppressing factor 9 RNA biogenesis leading to hemophilia B' to PLOS Genetics. Your manuscript was fully evaluated at the editorial level and by independent peer reviewers. The reviewers appreciated the attention to an important topic but identified some aspects of the manuscript that should be improved.

We therefore ask you to modify the manuscript according to the review recommendations before we can consider your manuscript for acceptance.

You will notice that while two of the reviewers had no additional comments, one reviewer has some remaining concerns that primarily focus on a few interpretive and organizational issues.  We do not believe any additional experimentation or major alterations will be required to address these comments.  However, we are hopeful you can address these points with a few minor revisions, and we are likely to solicit one final review from this reviewer before making a final decision.

[LINK]

Yours sincerely,

Gregory M. Cooper, PhD

Associate Editor

PLOS Genetics

Gregory Barsh

Editor-in-Chief

PLOS Genetics

Reviewer's Responses to Questions

**Comments to the Authors:**

Reviewer #1: I am satisfied that the authors have addressed the concerns I brought up.

Reviewer #2: the authors have addressed my concerns

Reviewer #3: The mns is significantly improved and the Authors have appropriately addressed most of the criticism raised also performing novel experiments. However there are still some points that needs attention.

1 In my opinion the Authors have now sufficient evidence to prove that the creations of a consensus 5’ss in the 3’ UTR reduces the total amount of mRNA through an U1snRNP-dependent inhibition of polyadenylation as well as to exclude that the variant induces aberrant splicing. The other potential mechanisms the Authors exclude, miRNA, RBPs and NMD, are not experimentally validated (miRNA and RBPs only in silico and NMD just because splicing is not involved). In particular, for NMD the minigene systems in general (and in particular the one used here, which has no introns) are not NMD sensitive. My suggestion is to focus, in the result section, on the experimentally validated mechanism first and then say that the other mechanisms, based on indirect evidence, are very unlikely.

2 The Authors have correctly analyzed the effect of the suppressor U1 on the WT mutant clearly showing that it significantly reduces the amount of mRNA. I believe this is an important and critical experiment that prove the direct involvement of U1 and accordingly it could be inserted in the main figures. (If there is no space one of the two ASO approaches that block the U1 snRNP should go into supplementary). However, I feel that there is a mistake in the data interpretation at page 10 lanes 270-273 concerning the effect of suppressor U1s on the mutant minigene. I do not see an increase in RNA expression after transfection of the suppressor U1 and I do not understand why a “stronger F9 mutant mRNA increase” should be expected in this case. The Authors should note that the suppressor U1 makes a good complementarity also with the mutant sequence through a wobble G-U base pairing, thus I would expect a further reduction in the mRNA. In any case, the suppressor U1 experiment are not designed to block the U1 snRNP and thus this experiment should be presented in a separate result section.

3 Unfortunately, the half-life experiments do not prove a co-transcriptionally decay of the mutant F9 mRNA. The mutation clearly reduces but not completely disrupt the usage of the PAS, producing a low amount of normal mature F9 mRNA which has, as expected, a normal half-life. Thus, in the absence of more direct measurement (like Chip assay with PolII or nascent nuclear RNA) the conclusions should be better explained/modified accordingly.

4 The ASO and the U7 for masking the mutated site seem to induce additional bands (Fig 5 B and D) which are considered to originate from the use of downstream PASs. The Authors speculate in discussion that the ASO might prevent binding of a cellular factor. In the absence of more defined prove of their origin and considering that they are using hybrid minigenes, I suggest to be more cautious in their interpretation. I also see another unclear band in mutant untreated.

**Have all data underlying the figures and results presented in the manuscript been provided?**

Reviewer #1: Yes

Reviewer #2: Yes

Reviewer #3: Yes

PLOS authors have the option to publish the peer review history of their article (what does this mean?). If published, this will include your full peer review and any attached files.

Reviewer #1: No

Reviewer #2: No

Reviewer #3: No

---

## [Editor Report · Decision Letter 2]

22 Feb 2020

Dear Dr Bohne,

We are pleased to inform you that your manuscript entitled "Pathological mechanism and antisense oligonucleotide-mediated rescue of a non-coding variant suppressing factor 9 RNA biogenesis leading to hemophilia B" has been editorially accepted for publication in PLOS Genetics. Congratulations!

Yours sincerely,

Gregory M. Cooper, PhD

Associate Editor

PLOS Genetics

Gregory Barsh

Editor-in-Chief

PLOS Genetics

Comments from the reviewers (if applicable):

**Data Deposition**

http://datadryad.org/submit?journalID=pgenetics&manu=PGENETICS-D-19-01514R2

**Press Queries**

---

## [Editor Report · Acceptance letter]

23 Mar 2020

PGENETICS-D-19-01514R2 

Pathological mechanism and antisense oligonucleotide-mediated rescue of a non-coding variant suppressing factor 9 RNA biogenesis leading to hemophilia B 

Dear Dr Bohne, 

We are pleased to inform you that your manuscript entitled "Pathological mechanism and antisense oligonucleotide-mediated rescue of a non-coding variant suppressing factor 9 RNA biogenesis leading to hemophilia B" has been formally accepted for publication in PLOS Genetics! Your manuscript is now with our production department and you will be notified of the publication date in due course.

With kind regards,

Jason Norris

PLOS Genetics

On behalf of:
